# An Efficient Privacy Protection Mechanism for Blockchain-Based Federated Learning System in UAV-MEC Networks

**DOI:** 10.3390/s24051364

**Published:** 2024-02-20

**Authors:** Chaoyang Zhu, Xiao Zhu, Tuanfa Qin

**Affiliations:** 1School of Electronic and Information Engineering, South China University of Technology, Guangzhou 510641, China; zhucy@gxu.edu.cn; 2School of Computer and Electronic Information, Guangxi University, Nanning 530004, China; 3School of Electronic Information Engineering, Guangxi Vocational Technical Institute of Industry, Nanning 530001, China; zhuxiao@gxu.edu.cn; 4Guangxi Key Laboratory of Multimedia Communications and Network Technology, Guangxi University, Nanning 530004, China

**Keywords:** unmanned aerial vehicles, data privacy, federated learning, blockchain, poisoning attack

## Abstract

The widespread use of UAVs in smart cities for tasks like traffic monitoring and environmental data collection creates significant privacy and security concerns due to the transmission of sensitive data. Traditional UAV-MEC systems with centralized data processing expose this data to risks like breaches and manipulation, potentially hindering the adoption of these valuable technologies. To address this critical challenge, we propose UBFL, a novel privacy-preserving federated learning mechanism that integrates blockchain technology for secure and efficient data sharing. Unlike traditional methods relying on differential privacy (DP), UBFL employs an adaptive nonlinear encryption function to safeguard the privacy of UAV model updates while maintaining data integrity and accuracy. This innovative approach enables rapid convergence, allowing the base station to efficiently identify and filter out severely compromised UAVs attempting to inject malicious data. Additionally, UBFL incorporates the Random Cut Forest (RCF) anomaly detection algorithm to actively identify and mitigate poisoning data attacks. Extensive comparative experiments on benchmark datasets CIFAR10 and Mnist demonstrably showcase UBFL’s effectiveness. Compared to DP-based methods, UBFL achieves accuracy (99.98%), precision (99.93%), recall (99.92%), and F-Score (99.92%) in privacy preservation while maintaining superior accuracy. Notably, under data pollution scenarios with varying attack sample rates (10%, 20%, and 30%), UBFL exhibits exceptional resilience, highlighting its robust capabilities in securing UAV gradients within MEC environments.

## 1. Introduction

Unmanned aerial vehicles (UAVs) have emerged as a crucial innovation in wireless communication networks, offering significant benefits such as easy deployment, improved mobility, and direct connectivity with a clear line of sight. This technological advancement has sparked a notable increase in both academia and industry’s focus on UAV wireless communication networks. In this field, UAV-assisted Mobile Edge Computing network (UAV-MEC) has gained recognition as a transformative concept. MEC utilizes artificial intelligence (AI) to process the vast amount of data collected by widespread drone networks, enabling the provision of intelligent services [1]. However, deploying these edge computing networks in potentially hostile environments presents various security and privacy challenges. Innovative methods are crucial to safeguard data privacy, maintain model accuracy, and enable robust data processing auditability within the UAV-MEC network [2].

Federated learning (FL) emerges as a novel AI approach that utilizes decentralized data and training [3,4]. It empowers UAVs to leverage their locally collected data to build localized deep learning models. These models are then transmitted to a central node for aggregation, resulting in a global model. Ntizikira et al. [5] proposed the SP-IoUAV model, combining FL with CNN-LSTM networks to achieve both operational security and data privacy in the Internet of Unmanned Aerial Vehicles (IoUAV). This model outperforms previous approaches with its real-time anomaly detection and multi-factor authentication capabilities. Ref. [6] explores a group signature-based algorithm for federated learning in FANETs, highlighting its ability to safeguard node identities, minimize communication overhead, and improve security and privacy.

However, existing FL approaches in UAV-MEC networks face security and privacy risks due to the large number of UAVs and need for real-time response [7,8]. The central curator, which aggregates insights from distributed UAV nodes, is often a primary target for cyber-attacks, jeopardizing the integrity and confidentiality of the collective learning process [9]. Moreover, the system’s reliance on accurately recording contributions from diverse UAVs introduces vulnerabilities, as malicious entities can manipulate or falsify their contributions, resulting in skewed or compromised learning outcomes [6].

Blockchain technology offers a promising solution by enabling secure and decentralized data sharing, mitigating central server vulnerabilities, and facilitating tamper-proof record keeping of transactions through its immutability and auditability features [10]. This paves the way for enhanced security and privacy in collaborative learning within UAV-MEC networks. Ref. [11] proposes FedEx, a novel FL framework that utilizes mobile transporters to establish indirect communication channels between server and clients, achieving convergence in both synchronous and asynchronous versions.

Nevertheless, deploying blockchain-based FL (BFL) in UAV-MEC networks confronts various hurdles, including limited computational resources on UAVs, potential scalability issues with large numbers of participants, and inherent trade-offs between security and performance [12]. In certain fields, like healthcare, the integration of BFL is further complicated by the limited availability of data from various sources, such as hospitals and clinics [13]. Furthermore, the Internet of Things (IoT) environment presents its own unique set of challenges, including concerns regarding security and privacy [14,15]. Another challenge in federated learning is ensuring the quality of local training data, as there is no control over the data used for training.

Several studies address BFL challenges, such as secure aggregation or data encryption [16,17]. For instance, Mrad et al.’s proposed federated learning framework for UAVs focuses on addressing energy constraints and class imbalance, critical factors for UAV swarm performance, but it does not delve into the broader security implications of BFL [18]. Similarly, the SFAC framework by Wang et al. utilizes blockchain for secure data exchange and local differential privacy for user privacy, while incorporating an incentive mechanism [19]. SFAC’s effectiveness in fully decentralized settings with highly skewed or non-IID data distributions remains a potential concern. In [20], the author designs a privacy-preserving byzantine-robust federated learning (PBFL) scheme based on blockchain that uses cosine similarity to judge the malicious gradients uploaded by malicious clients and adopts fully homomorphic encryption to provide secure aggregation. Utilizing fully homomorphic encryption and cosine similarity for identifying malicious gradients can introduce significant computational overhead, potentially limiting the scheme’s applicability in real-time or resource-constrained scenarios. Building upon differential privacy success [21], Xu et al. [22] propose VerifyNet, a privacy-preserving and verifiable framework that leverages differential privacy’s noise-adding mechanism to protect individual data while allowing users to verify the integrity of the aggregated model and detect malicious updates. Ref. [23] evaluates the practical benefits of applying federated learning with local differential privacy in a real-world setting.

However, existing differential privacy federated learning methods often focus on the technical aspects, overlooking the broader context of balancing privacy and accuracy in real-world applications. For instance, existing empirical methods that rely solely on differential privacy to protect user data often struggle to find an ideal balance between privacy and model accuracy [24]. This makes them unsuitable for practical applications that require both privacy and performance. Moreover, optimizing differential privacy parameters remains a challenge in dynamic UAV-MEC environments characterized by resource constraints and potential data collection attacks [25]. This limited scope results in incomplete solutions that fail to address real systemic problems or lack versatility.

Therefore, our investigation focuses on the examination of two primary domains: (1) algorithms for automatic adjustment of parameters, specifically those that rely on privacy budgets or adversarial training, and (2) approaches to identify and alleviate specific forms of attacks such as poisoning attacks, data injection, and model manipulation.

### 1.1. Motivations and Contributions

In summary, implementing blockchain-based federated learning (BFL) in UAV-MEC networks holds immense potential, but faces several key issue that require thoughtful solutions:Privacy Exposure: Uploading local model parameters poses a privacy risk, as compromised edge servers could exploit them to access sensitive user data. Robust encryption techniques and secure communication protocols are crucial to mitigate this risk.Malicious Local Training: Malicious actors may attempt to manipulate the learning process through poisoned data or poor-quality datasets, compromising the global model’s integrity. Robust anomaly detection mechanisms and data quality checks are essential safeguards.Privacy-Preserving Trade-offs: Techniques like differential privacy and secure multi-party computation offer valuable privacy protection, but may introduce trade-offs in model accuracy or training efficiency. Finding the optimal balance between privacy and performance requires further research and development.

Motivated by these challenges, we propose a novel blockchain-based privacy protection method for federated learning (UBFL). Our goal is to provide robust safeguards for individual data while enabling efficient collaborative learning. This paper makes the following key contributions:We presents a novel, blockchain-based framework (UBFL) for privacy-preserving federated learning in UAV-MEC networks. Addressing the limited computing power of individual drones, UBFL leverages secure and decentralized parameter aggregation via blockchain smart contracts, significantly mitigating risks associated with centralized services.Furthermore, an innovative adaptive nonlinear function encryption algorithm is proposed to ensure robust gradient protection. This algorithm dynamically learns hierarchical constraints through fine-grained parameters, effectively addressing the challenges of manually selecting differential privacy parameters.To further enhance data security, a novel anomaly detection protocol utilizes the Random Cut Forest algorithm to identify and filter out potentially malicious gradients, ensuring the integrity of the model update process.Extensive experiments on the CIFAR10 and MNIST datasets demonstrate the effectiveness of the proposed encryption algorithm, particularly its outstanding resilience against data poisoning attacks up to 30%. This showcases its potential as a transformative solution for securing UAV-MEC networks.

### 1.2. Paper Organization

The remainder of this paper is organized as follows. Section 2 reviews related work, drawing comparisons with existing privacy and security solutions in UAV-MEC networks to underscore the need for our proposed methodology. The UBFL model design and scheme formulation, demonstrating the foundational elements of our approach, are outlined in Section 3. Section 4 details the methodology, elaborating on the design and implementation. An adaptive nonlinear function-based algorithm and the use of Random Cut Forest (RCF) for anomaly detection algorithm are proposed. The outcomes of the simulation are presented in Section 5. In Section 6, the study’s limitations are discussed. Finally, we summarize the key findings and their implications for the development of more secure and efficient UAV-MEC networks in Section 7.

## 2. Related Works

### 2.1. UAV-Enabled Mobile Edge Computing

Mobile Edge Computing (MEC) signifies a transformative shift in cloud computing, strategically situating computing and storage resources within the radio access network. This paradigm is instrumental in propelling applications, data, and services proximally to mobile users, thereby offering substantial reductions in latency, enhanced location awareness, and alleviated network congestion. This approach marks a significant departure from the traditional centralized cloud services, introducing a new dimension of efficiency and responsiveness in mobile computing. The integration of MEC nodes at the edge of UAV networks, as comprehensively analyzed in [26], proposes a UAV-assisted MEC offloading scheme, specifically designed to minimize task completion time for computation-intensive IoT tasks. Furthermore, Refs. [27,28] have developed a mobility-aware caching scheme within UAV networks enabled by MEC. This scheme is meticulously tailored to optimize content placement, trajectory planning, and bandwidth allocation, thereby minimizing latency and enhancing overall network performance.

### 2.2. Privacy Preserving of Federated Learning for Wireless Nework

Federated Learning emerges as a cutting-edge distributed machine learning approach, wherein participants engage in training local data and subsequently upload updated parameters to a centralized server for aggregation [29,30]. This innovative approach not only enhances learning efficiency but also effectively resolves the challenges of data silos and fortifies local data privacy, thereby representing a significant advancement over traditional machine learning paradigms. In contemporary neural network models, gradient descent is employed for parameter updates. However, this process poses a risk, as the exposure of participant gradients can inadvertently lead to the leakage of sensitive network parameters [31,32]. In [33], the paper proposes a channel-aware distribution and aggregation scheme to enforce equal contribution from all devices in the FL training as a means to resolve the global bias problem of aerial FL in large-scale UAV networks.

Differential privacy emerges as a pivotal concept designed to quantify and mitigate the risks associated with personal information exposure. It provides a robust privacy framework, employing sophisticated randomization techniques. The integration of differential privacy mechanisms within federated learning perturbs model parameters, thus safeguarding users’ private training data while still enabling the collaborative training of an accurate shared model [34,35]. This strategic approach effectively addresses the privacy concerns that have been a significant impediment to the real-world deployment of federated learning systems. Ref. [36] proposes DPFed, a differential private federated learning algorithm using the moments accountant technique. This achieves tighter privacy guarantees while preserving high model utility. Ref. [37] develops a Laplace mechanism-based differential private algorithm for federated learning. This leverages the exponential mechanism to preserve user privacy in model training.

In summary, while existing studies demonstrate that differential privacy can facilitate privacy-preserving federated learning, there is a pressing need for more comprehensive evaluations that consider factors such as single points of failure. Moreover, the differential privacy algorithm faces significant challenges due to its over-reliance on empirical methods for the selection of differential parameters.

### 2.3. Blockchain-Enabled UAV Federated Learning

Blockchain technology, characterized by its decentralization, immutability, and distributed ledger features, functions as a digital transaction ledger that is replicated and shared across network nodes, thereby eliminating the necessity for a central authority.

Its applicability in UAV scenarios is particularly highlighted by these inherent features. Ref. [38] proposes a blockchain-based incentive mechanism for UAV networks using a privacy-aware auction and consensus algorithm. This approach introduces a privacy-respecting reward mechanism to stimulate participation. Ref. [39] develops a distributed path planning and target tracking algorithm for UAVs using smart contracts on blockchain. This preserves participants’ privacy while enabling real-time path optimization in a collaborative manner.

Overall, these studies underscore the advantages of blockchain in enhancing UAV privacy, security, and reliability. However, to validate their applicability in real-world scenarios, larger-scale experiments that consider practical constraints, such as energy consumption and flight dynamics, are essential. Additionally, there is a need for an in-depth analysis of the optimized trade-offs between privacy/security and energy efficiency to further solidify these findings, as will be discussed in the subsequent section.

### 2.4. Anomaly Detection Using Random Cut Forest

Random Cut Forest (RCF) is an advanced unsupervised algorithm designed for anomaly detection within datasets, identifying data points that significantly deviate from established patterns or structures [40,41]. Anomalies, such as unexpected spikes in time series data or atypical data points, can drastically increase the complexity of machine learning tasks [42].

RCF assigns an anomaly score to each data point, where low scores denote normality and high scores indicate the presence of anomalies. The determination of these scores is application-specific, but typically, scores exceeding three standard deviations from the mean are considered anomalous. RCF’s adaptability extends to handling multi-dimensional input, setting it apart from many algorithms that are confined to one-dimensional time series data. Amazon SageMaker’s implementation of RCF demonstrates effective scalability with respect to the number of features, dataset size, and the number of instances.

The fundamental principle of RCF involves constructing a forest of trees, each originating from a partition of a sample of the training data. For instance, a random sample is divided according to the number of trees in the forest, with each tree organizing its subset of points into a k-d tree. The anomaly score for a data point is determined by the expected change in the tree’s complexity upon incorporating that point, inversely proportional to the point’s depth in the tree. RCF calculates an anomaly score by averaging the scores from each constituent tree and scaling the result in relation to the sample size.

## 3. System Model and Threat Analysis

### 3.1. Federated Optimization Model

We consider federated optimization problems as follows.
(1)$$\begin{array}{c} \min_{x\in R^{d}}\left[F\left(x\right):=\frac{1}{m}\sum_{i=1}^{m}F_{i}\left(x\right)\right], \end{array}$$
where *m* is the number of local models (clients) and $F_{i}\left(x\right)=E_{\xi_{i}∼D_{i}}\left[F_{i}\left(x,\xi_{i}\right)\right]$ is the local objective function associated with local data distribution $D_{i}$.

Typically, traditional federated learning comprises multiple participants and a server component, as illustrated in Figure 1. In this framework, participants train shared models, after which the server aggregates these local models and distributes tasks to the participants. The federated learning training process can be delineated into three steps:

Step 1: Task initialization and model BroadcastPrior to training, the server initially defines the tasks and objectives of the training session. It then selects devices for participation in federated learning and dispatches the shared model to these chosen devices.Step 2: Local training and updatesAt each communication round *t*, each client *k* trains a local model $M_{k}$ on its dataset $D_{k}$. The local update is represented as:
(2)$$w_{k}^{\left(t+1\right)}=w_{k}^{\left(t\right)}-\eta \nabla F_{k}\left(w_{k}^{\left(t\right)}\right)$$
where $w_{k}^{\left(t\right)}$ represents the model weights at iteration *t*, η is the learning rate, and $\nabla F_{k}\left(w_{k}^{\left(t\right)}\right)$ is the gradient of the loss function $F_{k}$ computed on $D_{k}$. The loss function $F_{k}\left(w\right)$ for each UAV client could be the cross-entropy loss for classification tasks, defined as:
(3)$$F_{k}\left(w\right)=-\sum_{\left(x,y\right)\in D_{k}}y\log\left(f_{w}\left(x\right)\right)+\left(1-y\right)\log\left(1-f_{w}\left(x\right)\right)$$
where (x,y) are the data samples and their labels, and $f_{w}\left(x\right)$ is the model’s prediction. Here, the local loss function can be different for different FL algorithms [28]. For example, with a set of input–output pairs ${\left\{x_{i},y_{i}\right\}}_{i=1}^{K}$, the loss function F of a linear regression FL model can be defined as $F\left(w_{k}\right)=\frac{1}{2}\left(x_{i}^{T}w_{k}-\right.$${\left.y_{i}\right)}^{2}$. Then, each client *k* uploads its computed update $w_{k}$ to the server for aggregation.Step 3: Global model aggregationAfter local training, clients send their model updates $w_{k}^{\left(t+1\right)}$ to the central server. The aggregation of local model updates to the global model on the blockchain is represented as:
(4)$$w_{G}^{\left(t+1\right)}=\frac{1}{\sum_{k\in K}\left|D_{k}\right|}\sum_{k=1}^{K}\left|D_{k}\right|w_{k}^{\left(t+1\right)}$$
where *k* is the total number of clients, $n_{k}$ is the number of samples on client *k*, and $w_{k}^{\left(t+1\right)}$ represents the parameters of the model updated by client *k*. We solve the following optimization problem:  
(5)$$\min_{w_{i\in K}}\frac{1}{K}\sum_{i=1}^{K}F\left(w_{i}\right)$$
subject to (C1): $w_{1}=w_{2}=\cdots w_{i}=w_{G}$.In this context, the loss function F serves as an indicator of the federated learning (FL) algorithm’s accuracy, such as in an FL-based object classification task. The constraint (C1) ensures uniformity in the learning model among all clients and the server for each FL task after every training round.The optimization problem in Equation (Equation 5) is typically solved using a gradient descent approach. For federated learning, an iterative process is applied as follows:
(6)$$w_{G}^{\left(t+1\right)}=w_{G}^{\left(t\right)}-\eta_{G}\nabla F_{G}\left(w_{G}^{\left(t\right)}\right)$$
where $\eta_{G}$ is the global learning rate and $\nabla F_{G}\left(w_{G}^{\left(t\right)}\right)$ is the average gradient of the global loss function. The convergence of the global model can be shown by demonstrating that the loss function decreases over iterations:
(7)$$F_{G}\left(w_{G}^{\left(t+1\right)}\right)\leq F_{G}\left(w_{G}^{\left(t\right)}\right)$$

Following the model’s derivation, the server disseminates the updated global model parameters $w_{G}$ to all clients. This dissemination is crucial for refining the local models in the subsequent learning round. The FL process is repeated iteratively until the global loss function stabilizes or a predetermined level of accuracy is attained.

### 3.2. Threat Models and Design Goals

In federated learning, significant advancements have been made in enhancing learning efficiency, resolving data silos, and protecting local data privacy. However, this progress is accompanied by inherent vulnerabilities. Local UAV nodes are particularly prone to privacy breaches, while base station edge nodes face substantial challenges in effectively identifying trustworthy local UAV nodes and mitigating sophisticated malicious attacks.

Within the federated learning framework, where multiple edge nodes collaborate to develop a global model, there are heightened risks posed by malicious users or edge nodes exploiting system vulnerabilities for personal gain, as shown in Figure 2. These risks include unauthorized access to model parameters and the uploading of inaccurate or substandard local model parameters, potentially undermining the integrity and effectiveness of the global model. This study focuses on mitigating specific threats, including:Privacy Leakage: Despite federated learning’s design, which involves transmitting only model parameters and not raw data, recent advancements in privacy attack methodologies have shown that adversaries can deduce sensitive information about local device data by analyzing these parameters.Poisoning Attack: The federated learning process is vulnerable to disruptions caused by malicious devices. These devices can compromise the process by tampering with raw data or submitting intentionally falsified local gradients, thereby threatening the accuracy and reliability of the global model.Single Point of Failure Attack: A critical vulnerability in federated learning is its reliance on a central server. If this server is compromised, the entire training process could be disrupted, leading to significant operational challenges.

In response to these identified threats, the study proposes a comprehensive algorithm that adheres to design objectives, focusing on privacy, accuracy, and resilience to attacks:Privacy Preservation: The algorithm is designed to protect user data privacy throughout the federated learning process. It specifically safeguards sensitive information within the model parameters uploaded by UAVs, preventing unauthorized access by malicious edge nodes. By integrating advanced privacy-enhancing techniques, the algorithm ensures secure transmission and storage of UAV model parameters, upholding user privacy.Model Accuracy Preservation: The algorithm anticipates and counters potential threats from malicious drones submitting corrupted or manipulated model parameters. It aims to prevent poisoning attacks that could degrade the global model’s accuracy. Incorporating robust validation mechanisms and data integrity checks, the algorithm ensures that privacy preservation does not compromise model accuracy.Resilience to Single-Point Attacks: Recognizing the susceptibility of federated learning systems to single-point failures, the algorithm employs blockchain technology’s collective maintenance features. Utilizing smart contracts and distributed ledger systems, it decentralizes the parameter aggregation process, enhancing the system’s resilience and providing a transparent and auditable training process.

### 3.3. UBFL Training Process

Recent advancements have led to the success of blockchain and federated learning algorithms within the drone sector, tackling the privacy protection challenges in collaborative training and data exchange among drone clusters. In this context, this paper introduces an approach known as Blockchain-enabled Federated Learning UAV Mobile Edge Computing (UBFL) network. This network integrates the foundational principles of blockchain and federated learning to ensure comprehensive data privacy protection for drones.

The UBFL architecture represents a seamless fusion of blockchain and federated learning principles, engineered to facilitate secure and privacy-preserving collaboration. Subsequent sections will offer an in-depth analysis of the UBFL architecture and a detailed delineation of the UBFL training workflow. Figure 1 visually depicts the proposed architectural construct.

#### 3.3.1. Network Model

The UBFL system architecture is illustrated in Figure 3. The architecture comprises a multi-UAV-assisted air–ground Mobile Edge Computing (MEC) network, consisting of *K* UAVs and *M* Base Stations (BSs). The UAV set is denoted as K≜1,2,…,K, and the BS set as M≜1,2,…,M. Given the UAVs’ inherent limitations in battery life and computing capabilities, they are not inherently suited for efficiently undertaking resource-intensive tasks. It is thus assumed that BSs are equipped with MEC servers, which are tasked with providing computing services to UAVs. The network utilizes blockchain technology to create a decentralized federated training platform, ensuring the secure storage of private data. The global server, enhanced by MEC, is designed to address the computational constraints of UAVs. The UBFL network is structured into two layers: the user layer, consisting of UAV mobile terminals, and the edge service layer, encompassing base stations with MEC servers that provide storage and computing capabilities. The MEC server is responsible for the calculation and updating of global model parameters. Ultimately, the UBFL network integrates the capabilities of blockchain and federated learning (FL) with the support of MEC servers. In this setup, blockchain provides a decentralized training platform, while MEC servers address the computational limitations of UAVs and aid in the computation of the global model.

UBFL encompasses three primary entities: UAVs, base station edge nodes, and the blockchain network.

Local Drone. These are drone devices situated at the network’s edge, equipped with limited local datasets and computing capabilities. Their objective is to construct a more accurate machine learning model through federated learning in collaboration with other drone-based devices. This approach aims to provide smarter services while simultaneously safeguarding data privacy.MEC edge node. Miners, integral to the blockchain network, are typically equipped with substantial computing and communication resources. These resources enable them to provide essential services such as validation, consensus building, and other critical functions within the blockchain infrastructure.Blockchain network. The blockchain network plays a pivotal role in managing the registration of users and base station edge nodes, as well as in the aggregation of global models, thereby serving as a fundamental component in the orchestration of the system’s overall functionality.

#### 3.3.2. UBFL Training Process

In the UBFL, each drone executes computations and exchanges training updates via a blockchain ledger on the edge network. This approach facilitates direct global model aggregation on local devices, thus obviating the need for a central server. The blockchain service, operational on the MEC server, is responsible for receiving, storing, and authenticating UAV-uploaded model parameters through consensus protocols. Furthermore, UBFL effectively mitigates the network latency issues commonly associated with central server communications [43]. The UBFL system’s training process is illustrated in Figure 4.

Registration: MEC server Miners and UAV devices apply for registration with the task publisher, providing details including the size of their local datasets, client ID, hash code, and reward.Local Training and Encryption: UAVs train the machine learning model on their local datasets, performing $n_{i}$ iterations on the obtained gradient, then they compute the shared parameter gradient per several batches trained and encrypt the obtained gradient with adaptive nonlinear function to deal with threat 1.Transmission of Encrypted Data: The drone sends the encrypted gradient and digital signature to the associated miner base station edge node in a blockchain transaction format. Privacy of the model is guaranteed by incorporating a non-linear function into the gradients, as specified in Algorithm 1.Data Verification: Upon receiving the data, miners undertake the task of authenticating the signature in order to safeguard against any potential alteration or tampering. Specifically, gradients recognized as normal through the utilization of the RCF Algorithm 2 are subject to multiplication by the present cumulative reward in order to ascertain the cumulative probability prior to the selection of the drone.Legitimacy Verification and Global Weight Aggregation: The verification committee selects an edge node through mining voting to create a new block, and uses the hash of the edge node to calculate the data digest. Due to attempting to tamper with the content of the block, all blocks before it need to be rewritten, otherwise the blockchain will be broken The operation of rewriting blocks requires enormous computing resources, ensuring the immutability of local ledgers at each node.Model Update and Training Continuation: The UAVs download the new block from their associated miner, extract the global gradient for local model updates, and initiate the subsequent training round, beginning again from Step 2. This cycle continues until the model converges or reaches the maximum number of training rounds.

**Algorithm 1:** Adaptive Nonlinear Privacy Protection Algorithm.
**Input**: number of UAVs $N_{U}$, number of base stations $N_{B}$, number of global rounds $R_{0}$, minimum number of participating UAVs $P_{r}$
**Output**: Final global model after $R_{0}$ rounds of training

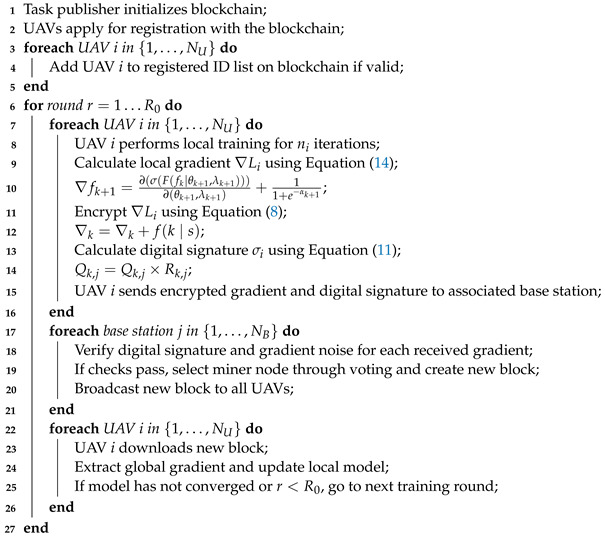



This procedure provides a thorough depiction of the UBFL system, highlighting the integration of blockchain technology to enhance security and effectiveness in federated learning environments. Further elaboration is provided in the subsequent section.
**Algorithm 2:** Identity Authentication and Real-Time Gradient Detecting algorithm.**Input**: number of UAVs $N_{U}$, number of base stations $N_{B}$, minimum number of participating UAVs $P_{r}$**Output**: Enhanced system efficiency and security
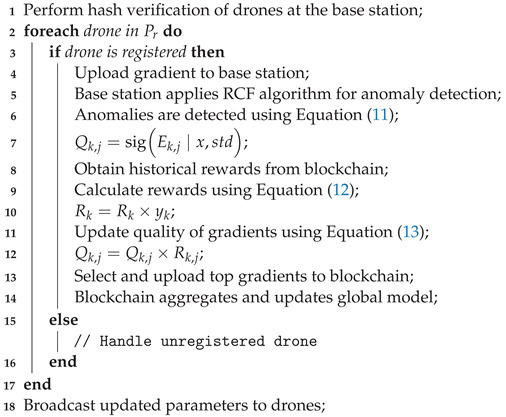


## 4. Algorithm Design and Solution

To enhance the trustworthiness and resilience against poisoning attacks within the UBFL network, our proposed algorithm is designed to meet critical objectives: preservation of privacy, maintenance of model accuracy, and protection against single-point attacks. This algorithm integrates privacy-enhancing techniques and leverages the robust capabilities of blockchain technology, thereby ensuring the protection of user privacy, the maintenance of model accuracy, and the enhancement of the federated learning system’s robustness against a diverse array of threats.

### 4.1. Adaptive Nonlinear Privacy Protection Algorithm for UAV Local Training

Recent research has highlighted the challenge of selecting differential parameters in differential privacy algorithms. To tackle this, researchers have developed various innovative methods, including automated tuning, robust estimation techniques, theoretical bounds, and adaptive mechanisms.

Advancing these methodologies, our study introduces an adaptive function-based algorithm specifically designed to protect the privacy of uploaded UAV gradients, as shown in Algorithm 1. This algorithm utilizes a sigmoid function for the nonlinear transformation of adaptive parameters across various layers. Such an approach ensures that the adaptive parameters from different layers cumulatively contribute to the current layer, thereby circumventing the training oscillation problem commonly encountered when optimizing parameters for each layer independently. The local nonlinear function employed in the UBFL algorithm is explicated in Equation (Equation 8).
(8)$$\left\{\begin{array}{c} \nabla_{k}=\nabla_{k}+f\left(k|s\right)\\ S\leftarrow \left(\alpha_{1},\cdots \alpha_{N}\right),N\mathrm{is}\mathrm{the}\mathrm{shared}\mathrm{layers}\; \end{array}\right.$$

In the equation, *S* represents the set of adaptive parameters corresponding to each layer of the shared network, where parameter *k* denotes the *k* layer of the shared network. The term $\alpha_{k}$ signifies that the initial value of the adaptive parameter for the *k* shared network layer is set to one. The parameter *N* indicates the total number of shared layers within the network. The function f(k∣s) is defined as an adaptive nonlinear encryption function. For an exhaustive exposition of detailed equations, calculations, and theoretical analysis supporting the efficacy of the adaptive nonlinear encryption function, refer to Appendix A. This appendix substantiates the claims made regarding the function’s advantages. The complete expression of the noliner activation function is presented below.
(9)$$\left\{\begin{array}{c} f\left(k|s\right)=\frac{1}{1+e^{-\alpha_{k}}}\\ ∥\alpha_{k}∥=1,\alpha_{k}\leq 1 \end{array}\right.$$

Equation (Equation 9) provides a detailed mathematical representation of the adaptive nonlinear encryption function utilized in the UBFL system. This function, f(k∣s), is defined as a sigmoid function, where it represents the adaptive parameter corresponding to the *k* layer of the shared network. The equation stipulates that the initial value of $\alpha_{k}$ for each layer is set to one, and it is constrained to remain at or below this value throughout the training process. This constraint ensures that the adaptive parameters do not exceed a predefined threshold, thereby maintaining stability and consistency in the training process.

The parameters of the shared layers are not static but dynamically adjust in response to the progression of the local neural network training. This dynamic adjustment is crucial for aligning the shared layer parameters with the evolving training process, ensuring that they effectively contribute to the overall learning objective. This integration is a key aspect of the training methodology, as it allows the shared layer parameters to directly influence and refine the classification accuracy of the local neural network.

Consequently, the local loss function of the UAV is bifurcated into two primary components. The first component encompasses the traditional aspects of neural network training loss, while the second component is uniquely characterized by the inclusion of the adaptive parameters from the shared layers. This dual-component structure of the local loss function is a novel approach in federated learning, particularly in the context of UAV applications, where it addresses the specific challenges and requirements of UAV-based neural network training.

In the context of an image classification dataset, this study categorizes private datasets into two distinct types: local datasets and block datasets. When creating a new block, the dataset of the old block is replicated to the new block. The overarching goal of the entire training process is to minimize the classification loss associated with training the image classification dataset. For this purpose, the classification loss is calculated using the cross-entropy loss method, which is widely recognized for its effectiveness in such tasks.The nonlinear disturbance loss attributed to adaptive parameters at each layer of the network is conceptualized as a nonlinear regularization term. This approach to loss calculation introduces an additional layer of complexity and refinement to the training process. This loss function is integral to the training process, as it ensures that the adaptive parameters contribute effectively to the overall learning objective while maintaining the stability and robustness of the model.


(10)
$$\left\{\begin{array}{c} {Loss}_{k}=G-\sum_{m=0}^{B}\sum_{n=0}^{C}p_{j}\times \log\left(p_{j}\right)\\ G=\sum_{m=0}^{N}e^{\sum_{k=0}^{s}abs}\left(\alpha_{k}\right) \end{array}\right.$$


In Equation (Equation 10), $Loss_{k}$ represents the loss function for the *k* drone client in the UBFL system. The equation delineates two primary components: the cross-entropy loss and the nonlinear regularization term *G*. Here, *B* denotes the batch size, and *C* represents the number of categories in the image classification task. The cross-entropy loss, calculated as the sum of the product of the probability $P_{j}$ of each category *j* and its logarithm, is a standard approach in classification tasks for quantifying the difference between the predicted and actual distributions.

The term *G*, as defined in the equation, represents the nonlinear regularization term associated with the adaptive parameters of each layer. This term is computed as the sum of the exponential functions of the cumulative adaptive parameters $a_{a}$ up to the *m* layer, where *N* is the total number of layers. The inclusion of *G* in the loss function introduces a nonlinear aspect to the regularization process, enhancing the model’s ability to generalize and preventing overfitting. This nonlinear regularization is particularly crucial in the context of federated learning, where the model needs to be robust and adaptable to diverse and decentralized datasets.

In the UBFL system, the adaptive function encryption method offers several key advantages over traditional differential privacy:Fine-Grained Layered Adaptive Parameters: The UBFL system’s internal network for each UAV consists of a complex convolutional neural network, characterized by varying convergence speeds across its layers. To address this, the study establishes unique adaptive parameters for each shared network layer, allowing for tailored adaptation to their respective convergence speeds. This method ensures efficient convergence by taking into account the distinct characteristics of each layer.Hierarchical Constraint in Adaptive Parameter Learning: In contrast to adaptive differential privacy, which generally sets adaptive parameters based on a broad gradient convergence logic, the UBFL system integrates these parameters directly into the local neural network training process. They form a part of the loss function for each UAV neural network. Consequently, the ongoing training of the local neural network influences the adaptive parameters of each layer, guiding them to converge with the local loss and ultimately reach an equilibrium. Furthermore, the optimization of parameters at each layer is intricately linked to and constrained by the local loss experienced at that specific layer. This hierarchical constraint ensures a more nuanced and effective optimization process, tailored to the specific requirements and dynamics of each layer within the neural network.These advancements in the UBFL system’s adaptive function encryption method signify a substantial progression in the approach to privacy preservation and model optimization in federated learning, especially for UAV applications.

### 4.2. Identity Authentication and Gradient Selection Mechanism Using Blockchain in UBFL

In the UBFL system, the base station initiates the process with a hash verification to ascertain the registration status of each drone. This crucial step effectively filters out unregistered drones, thereby ensuring that subsequent gradient anomaly detection is conducted exclusively on registered drones. Once verified, registered drones within a specific region upload their gradients to the base station (edge node). These uploaded gradients are then subjected to anomaly detection using the Random Cut Forest (RCF) algorithm. The procedural details of RCF-based anomaly detection are illustrated in Figure 5.
(11)$$\left\{\begin{array}{c} Q_{k,j}=\mathrm{sig}\left(E_{k,j}|x,std\right)\\ x=\frac{1}{W}\sum_{j=0}^{W}E_{k,j},std=\mathrm{sqrt}\left(\sum_{j=0}^{W}{\left(E_{k,j}-x\right)}^{2}\right)\\ \mathrm{sig}\left(E_{k,j}|x,std\right)=\left\{\begin{array}{c} 1,x-2\times std\leq E_{k,j}\leq x+2\times std\\ 0,\;\mathrm{otherwhise}\; \end{array}\right. \end{array}\right.$$

Equation (Equation 11) defines the RCF algorithm’s parameters. Here, θ represents the gradient of the *j* drone collected by the *K* base station, *x* is the average gradient, std is the standard deviation, and *W* is the number of gradients collected by the current base station. The function sig determines the anomaly status of the current gradient, marking anomalous gradients as zero and normal gradients as one. The gradients are then sorted in reverse order using the inverse ranking method to obtain the final sorting result. Subsequently, the base station requests gradient information from the blockchain, which responds with the cumulative historical rewards for the registered drones. The reward function is defined as shown in Equation (Equation 12).
(12)$$\left\{\begin{array}{c} R_{k}=R_{k}\times y_{k}\\ y_{k}=\frac{Q_{k,j}\times W}{\sum_{j=0}^{W}Q_{k,j}} \end{array}\right.$$

In Equation (Equation 12), $R_{k}$ denotes the cumulative historical rewards of the *k* registered drone, and *R* normalized represents the normalized instant reward. Based on Equation (Equation 11) and Equation (Equation 12), the base station recalculates the quality of each drone’s uploaded gradient, as delineated in Equation (Equation 13).
(13)$$\left\{\begin{array}{c} Q_{k,j}=Q_{k,j}\times R_{k,j}\\ \mathrm{sorted}\left(Q_{k}\right) \end{array}\right.$$

In Equation (Equation 13), gradients identified as normal based on the RCF algorithm are multiplied by the current cumulative reward to determine the cumulative probability before drone selection. The gradients are then sorted in reverse order, with the top two and three gradients selected and uploaded to the blockchain. The blockchain performs secure aggregation based on these gradients, trains the global network, and periodically broadcasts the global shared parameters to all drones, thereby enhancing the overall efficiency and security of the UBFL system. We designed the algorithm as Algorithm 2.

## 5. Experiment

### 5.1. UBFL DNN Structures

The fundamental architecture of the local training neural network, a critical component of the UBFL system, is illustrated in Figure 6. The diagram in Figure 6 illustrates a neural network architecture designed for sequence processing tasks that benefit from both spatial and temporal feature recognition in the UAV-MEC network. The architecture includes three convolutional blocks, each possibly consisting of a convolutional layer followed by batch normalization (BN) and an activation function (denoted by α with subscripts indicating different functions or parameters for each block). The first convolutional block employs a filter size of 2 × 2 with a stride of one for the convolution operation. The batch normalization is applied post-convolution followed by an activation function $\alpha_{1}$. The output from the convolutional blocks is fed into a bidirectional LSTM layer. The Bi-LSTM allows the network to process sequences in both forward and reverse directions, capturing context from both past and future data points within a sequence. The notation “Cell Hidden = 256” specifies that each LSTM cell in the layer has a hidden state vector of size 256, indicating the capacity of the cell to capture and retain information over time. The processed sequence data are output through the top of the diagram, where each element of the sequence is assigned a label. This is indicative of sequence labeling tasks where each timestep of the input data is classified or regressed to a corresponding label, common in applications like time-series anomaly detection.

The architecture leverages the strengths of both CNNs for feature extraction from the input data and LSTM for capturing the temporal dependencies within the sequence. This combined approach is particularly advantageous in scenarios where the input data are a sequence with rich, spatially and temporally relevant features. The feature output of a given layer, denoted as $f_{k}$, influences the gradient output of the subsequent layer $f_{k+1}$, which is mathematically expressed as: (14)$$\nabla f_{k+1}=\frac{\partial \left(\sigma \left(F\left(f_{k}|\theta_{k+1},\lambda_{k+1}\right)\right)\right)}{\partial \left(\theta_{k+1},\lambda_{k+1}\right)}+\frac{1}{1+e^{-\alpha_{k+1}}}$$

In this equation, *F* represents the formal structure of each neural network layer, while $\theta_{k+1}$ and $\lambda_{k+1}$ are the shared and private parameters, respectively, at layer k+1. As indicated in Equation (Equation 14), the adaptive factor $\alpha_{k}$ functions as the noise component for the gradient of each layer.

### 5.2. Allocation of Local UAV Training Dataset

In the context of an image classification dataset, this study categorizes private datasets into two distinct types: local datasets and block datasets. When creating a new block, the dataset of the old block is replicated to the new block. The overarching goal of the entire training process is to minimize the classification loss associated with training the image classification dataset. For this purpose, the classification loss is calculated using the cross-entropy loss method, which is widely recognized for its effectiveness in such tasks.

The algorithm is evaluated using datasets with MNIST and CIFAR10 in this study. These datasets, representative of medium-complexity data typically gathered by local devices, are also extensively used in various edge computing scenarios. The allocation of the UAV client datasets is detailed in Table 1.
(15)$$\left\{\begin{array}{c} UAV_{1}\cup UAV_{2}\cup \cdots \cup UAV_{5}=D\\ \sum_{k=0}^{5}\times \;p_{k}=1\\ C_{1}\cup C_{2}\cup \cdots \cup C_{5}=C\\ \forall C_{k}\leq C,k\leq 5 \end{array}\right.$$

In this configuration, the global trainer (blockchain) does not directly allocate the dataset. Instead, it updates the shared parameters using the gradient uploaded by the base station (edge node) and subsequently performs secure aggregation. The dataset assigned to each UAV adheres to the constraints specified in Equation (Equation 15).

As show in Equation (Equation 15), the dataset allocation process involves random sampling by local trainers, with the number of sample categories in each dataset being determined by the specific sampling procedure. This strategy ensures that no single trainer possesses samples of all categories, thereby promoting diversity and robustness in the training process.

### 5.3. Implementation Details

In the development of our UBFL system, we uniquely configure a network with a single Base Station/Edge Computing (BS/EC) server overseeing $N_{U}=50$ UAVs, a setting that reflects practical UAV operational scenarios. Distinctively, the UAVs’ computing frequencies, $\gamma_{i}$, for all $i=1,\ldots ,N_{U}$, are determined through a sampling process from the range $\left[10^{6},10^{8}\right]$ Hz, tailored to emulate real-world UAV computational capabilities. The application of $\kappa =7\times 10^{4}$ CPU cycles and $P_{i}=0.28$ Watt for all UAVs optimizes the balance between computational demand and energy efficiency. Our network configuration is specifically designed to test the model weight transmission delays (20 ms to 200 ms) over a 10 MHz bandwidth, addressing a critical challenge in UAV communications.

To validate our defense approach against data poisoning in federated learning tasks, we deliberately chose the MNIST and CIFAR-10 datasets for their diversity in image complexity, directly correlating to varied UAV image processing tasks. MNIST, with its 60,000 28 × 28 pixel grayscale images, and CIFAR-10’s 60,000 32 × 32 pixel color images offer a comprehensive test bed, reflecting a wide range of potential UAV visual processing scenarios. This selection is underpinned by a methodical evaluation to ensure the datasets’ applicability in simulating UAV-specific challenges, particularly in image classification tasks pertinent to UAV surveillance and reconnaissance missions. Each UAV’s data handling capability is capped at $|M_{r}|\;=1300$ samples, a constraint that further simulates real-world operational limitations.

This bespoke setup, alongside a critical comparative analysis with existing federated learning frameworks, distinctly positions our research within the UAV domain. It not only underscores our methodological and experimental rigor but also the adaptability of our proposed solution to the nuances of UAV operations. By elucidating these unique aspects, we aim to distinguish our work from prior studies, ensuring that our contributions to the UAV-MEC network domain are both clear and original.

The experimental hyperparameters are outlined in Table 2. The batch size was set to 64, the learning rate was established at 0.001, and the truncation loss was fixed at 100. The optimization function utilized was the AdamOptimizer, with nonlinear adaptive parameters designated as $\alpha_{1}$, $\alpha_{2}$, and $\alpha_{3}$. This setup facilitates a comprehensive evaluation of the algorithm’s performance across different datasets and under various parameter configurations.

To evaluate its performance, a series of comparative experiments were conducted on a GPU, providing valuable insights into its comparative advantages over existing algorithms. The hardware configuration used for these comparative experiments is detailed in Table 3.

The evaluation indicators—*Accuracy*, *F1-Score*, *Precision*, and *Recall-Score*—are standard metrics for assessing the performance of classification algorithms.
(16)$$\left\{\begin{array}{c} Precision\left(p\right)=\frac{TP}{TP+FP}\\ Recall\left(r\right)=\frac{TP}{TP+FN}\\ F1=\frac{2\times P\times R}{P+R}\\ Accuracy=\frac{TP+TN}{TP+FP+TN+FN} \end{array}\right.$$
where *TP* is true positive, *FP* is false positive, *FN* is false negative, *TN* is true negative; *P* and *R* are precision and recall, respectively. A true positive is predicted to be positive and is actually positive. The positive sample is successfully predicted to be positive. A false positive is predicted to be positive but is actually negative. The negative sample is incorrectly predicted to be positive. A true negative is predicted to be negative and is actually negative. The negative sample is successfully predicted to be negative. A false negative is predicted to be negative but is actually positive. The positive sample is incorrectly preducted to be negative.

### 5.4. Effect of Adaptive Parameter Setting for Training Accuracy

Based on Cifar10 and Mnist datasets, the convergence of the global loss and adaptive parameter training process of the UAV-BFL algorithm is shown in Figure 7, and the test result curves are shown in Figure 8 and Figure 9.

Table 4 presents the results of experiments conducted on two datasets, CIFAR10 and MNIST. It details the initial and convergence values of the UAV-BFL training loss and three adaptive parameters, namely $\alpha_{1}$, $\alpha_{2}$, and $\alpha_{3}$. From the data, it is evident that $\alpha_{3}$ exerts the most substantial influence on loss reduction, as indicated by its lowest convergence values across both datasets. This observation implies a correlation between the depth of the adaptive parameter layers and their efficacy in diminishing the loss following the algorithm’s convergence.

More specifically, the result reveals that as the adaptive parameter layers approach closer to the output layer of the neural network, their convergence values tend to decrease. This trend suggests that the closer the layer is to the output, the greater its impact on reducing the overall training loss. Such insights are instrumental in understanding the dynamics of adaptive parameters within the UAV-BFL training process and their role in optimizing neural network performance.

#### Effect of Adaptive Nonlinear Function on Utility–Privacy Trade-Off

In this paper, we compare the performance of our solution with its various modifications where one or more components ($\alpha_{1}$, $\alpha_{2}$, $\alpha_{3}$) are omitted. By comparing these different versions, it can demonstrate the impact and importance of each component on the overall performance of the algorithm, as shown in Table 5. We conducted six comparative experiments:UAV-BFL-Without-$\alpha_{i}$: These variants of the UAV-BFL algorithm lack an adaptive parameter layer *i*, where *i* corresponds to each *a* highlighted in the table (e.g., $\alpha_{1}$, $\alpha_{2}$, $\alpha_{3}$). This suggests that the algorithm uses multiple adaptive parameter layers, and the experiments are testing the impact of each layer’s removal on the overall performance.UAV-BFL-DP: This variant represents the use of the traditional differential privacy algorithm in the comparison. Differential privacy is a system for publicly sharing information about a dataset by describing the patterns of groups within the dataset while withholding information about individuals in the dataset. By comparing UAV-BFL with UAV-BFL-DP, the paper likely aims to show how their novel approach compares to the traditional differential privacy methods in terms of the evaluation indicators.UAV-BFL-None-RCF: Indicates that this particular model variant does not include the RCF algorithm component. Since RCF is often used for anomaly detection, its absence in this variant would show how much the RCF algorithm contributes to the performance of the UAV-BFL model.

The analysis indicates that contribution factors from different neural network layers exert varying degrees of influence on inter-layer interactions. Factors located closer to the output layer are more effective in enhancing the algorithm’s accuracy, while those positioned further away tend to diminish it.

As outlined in Equation (Equation 10), these inter-layer contribution factors undergo a nonlinear transformation through a sigmoid function and are subsequently treated as regularization terms within the global loss function.

The study’s primary focus is to explore the influence of these adaptive nonlinear parameters, originating from distinct network layers, on the overall accuracy of the model. The findings are detailed in Table 6. The data provide clear evidence of the benefits of adaptive parameters, demonstrating their role in enhancing model prediction accuracy on Cifar10 and Mnist datasets.

### 5.5. Comparison with Differential Privacy Algorithms

This accuracy analysis demonstrates that our algorithm, which utilizes an adaptive layered contribution factor, achieves superior accuracy in privacy protection compared to methods based on differential privacy (DP). The DP algorithm, particularly when based on the Laplacian mechanism, tends to induce oscillations in the noise value during the randomness calculation. In contrast, our algorithm activates the layered contribution factor using a sigmoid function as gradient noise. This approach ensures that all factors rapidly converge to small values during the training process, resulting in greater stability. Consequently, the parameter updates in our algorithm’s model are more consistent, leading to enhanced robustness and accuracy.

Table 7 reveals that, in comparison to other algorithms, our proposed algorithm exhibits a notable improvement in accuracy on the CIFAR10 dataset, with a maximum increase of 4.336% (F1 Score) and a minimum increase of 2.076% (Accuracy). On the MNIST dataset, the maximum accuracy enhancement is 1.124% (Precision), while the minimum is 0.810% (Accuracy). Note that DP is a method based on Laplace perturbation with parameters set as follows: $\sigma^{2}=0.25,b=1.0$. Through the above accuracy analysis, it can be seen that the proposed algorithm achieves higher accuracy in privacy protection using adaptive layered contribution factor compared with the privacy protection method based on difference privacy (DP). The differential privacy algorithm based on the Laplacian will cause the noise value to oscillate when calculating the randomness of the noise. Relatively speaking, the layered contribution factor of the algorithm in this paper is activated by a sigmoid function as gradient noise, and all factors converge quickly to a small value in the training process and can be more stable. Therefore, the parameter update of the algorithm model in this paper tends to be more stable and can achieve higher robustness and accuracy.

#### Effect of RCF-Based Anomaly Detection on Model Poisoning Attack

In this study, noise samples were generated by randomly altering the labels of samples, with the proportions of these noise samples set at 10%, 20%, and 30%, respectively. Utilizing these noise samples, a series of poisoning attack experiments were conducted. These experiments were performed on the CIFAR10 and MNIST datasets, and the results are depicted in Figure 10 and Figure 11. This approach allowed for a comprehensive assessment of the impact of noise levels on the robustness of the models against poisoning attacks.

From Table 8, the F1 metric of the algorithm proposed in this paper exhibits significant improvements over the comparison algorithm. Under a 10% poisoning attack on the Cifar10 dataset, the algorithm achieved the highest increase in F1 value, with a boost of 26.18%. Similarly, under a 20% poisoning attack, the F1 value increased by a maximum of 29.33%, and under a 30% poisoning attack, the F1 value increased by a maximum of 22.49%. On the Mnist dataset, the algorithm demonstrated a maximum F1 value increase of 9.41% under a 10% poisoning attack, 11.94% under a 20% poisoning attack, and 23.36% under a 30% poisoning attack.

From Table 9, it is evident that the Recall metric of the UAV-BFL algorithm proposed in this paper outperforms the comparison algorithm, as shown by the results. When subjected to a 10% poisoning attack on the Cifar10 dataset, the UAV-BFL algorithm achieved the highest increase in Recall value, with a boost of 28.32%. Under a 20% poisoning attack, the Recall value increased by a maximum of 27.46%, and under a 30% poisoning attack, the Recall value increased by a maximum of 22.97%. On the Mnist dataset, the algorithm demonstrated a maximum Recall value increase of 10.72% under a 10% poisoning attack, 14.58% under a 20% poisoning attack, and 22.05% under a 30% poisoning attack.

From the perspective of privacy protection, the adaptive nonlinear function privacy protection method proposed in this study exhibits that its adaptive parameters rapidly converge to a very narrow range during model training. This convergence pattern is distinct from traditional differential privacy (DP) methods. In the proposed method, the minimal variation in adaptive parameters allows the aggregated gradient to more closely align with the actual gradient.

In terms of detecting abnormal gradients, the incorporation of the Random Cut Forest (RCF) algorithm in this research proves to be highly effective. This enhancement significantly boosts the efficiency of the proposed algorithm in the gradient aggregation phase. Additionally, the robustness of the proposed algorithm is set to be further validated across a wider array of datasets, underscoring its applicability and reliability in various data environments.

## 6. Discussion

This study innovatively enhances privacy and security in UAV-MEC networks by employing an adaptive nonlinear function-based algorithm with Random Cut Forest (RCF) for anomaly detection. Our empirical investigation, utilizing CIFAR10 and MNIST datasets, validates the efficacy of these methodologies, showcasing not only improved accuracy but also a robust defense against data poisoning attacks. Such advancements underscore the potential of our proposed solutions in setting a new benchmark for privacy protection in UAV-MEC ecosystems.

However, the research does not come without its limitations. The exploration into the computational complexity and scalability of our blockchain-based strategy, particularly within expansive UAV networks, remains partially unexplored. This oversight marks a critical area for future inquiry, essential for understanding the practicality of deployment in larger, real-world scenarios. Furthermore, while our findings indicate a superior performance compared to traditional methods, a comprehensive comparison across a broader spectrum of existing solutions is necessary for a more conclusive validation of our approach’s effectiveness. Lastly, the practical implementation of our system within the dynamic and constraint-laden UAV operating environments warrants more detailed investigation to fully ascertain its real-world applicability and operational feasibility.

Addressing these limitations will not only enhance the robustness of our proposed model but also broaden the scope of its applicability, paving the way for more secure and efficient UAV-MEC networks.

## 7. Conclusions

This study has introduced a UBFL method to enhance privacy protection within UAV (Unmanned Aerial Vehicle)-MEC (Mobile Edge Computing) networks. Our innovative approach overcomes the inherent limitations of traditional UAV-MEC networks by leveraging blockchain technology, thus establishing a decentralized framework that secures the integrity of model updates and ensures data validation without the need for central servers. The main achievements of our research include the development of an adaptive algorithm that utilizes a non-linear function for robust privacy preservation of UAV model updates and the application of Random Cut Forest (RCF) algorithms for effective anomaly detection to mitigate the risks of malicious data attacks. These contributions mark a significant advancement in privacy and security measures beyond the capabilities of existing methods.

Acknowledging areas for future exploration, we have identified opportunities to enhance the algorithmic efficiency for gradient verification and to develop consensus protocols specifically designed for UAV edge computing contexts. Moving forward, our focus will be on refining these aspects to address the highlighted limitations. Moreover, we aim to extend our research to assess the scalability and resilience of our proposed UBFL method in more complex network scenarios. The empirical results demonstrated within this study highlight the robustness of our algorithm against data pollution attacks across diverse pollution ratios, showcasing its applicability in a wide array of settings.

## Figures and Tables

**Figure 1 sensors-24-01364-f001:**
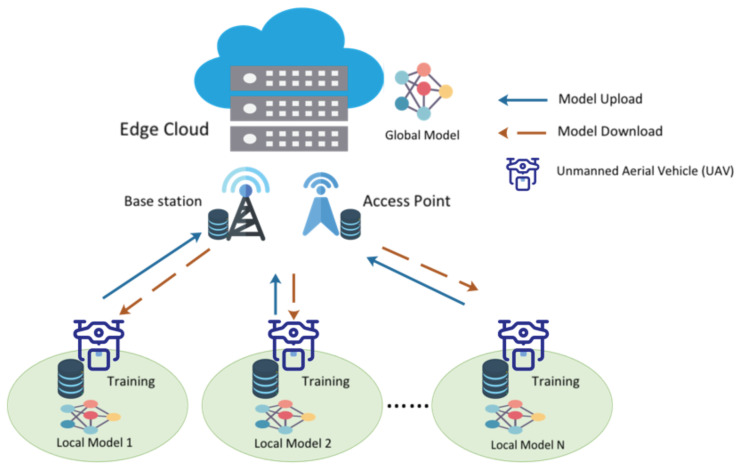
An example scenario of federated learning framework for UAV-assisted MEC.

**Figure 2 sensors-24-01364-f002:**
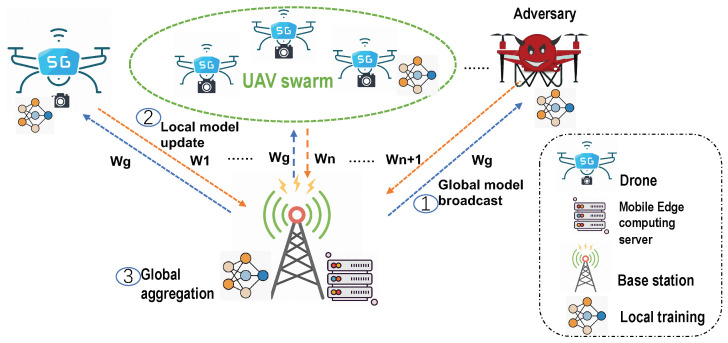
An FL training model with hidden adversaries who can eavesdrop trained parameters from both the clients and the server in UAV-MEC network.

**Figure 3 sensors-24-01364-f003:**
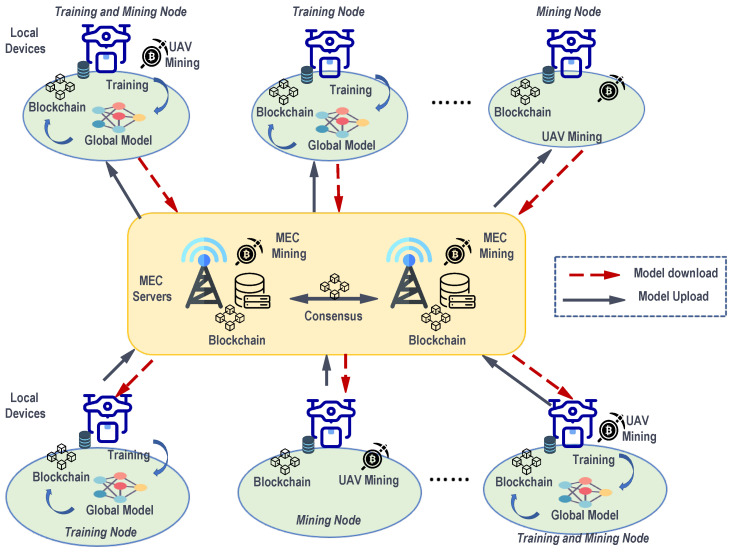
The conceptual design of the UBFL network.

**Figure 4 sensors-24-01364-f004:**
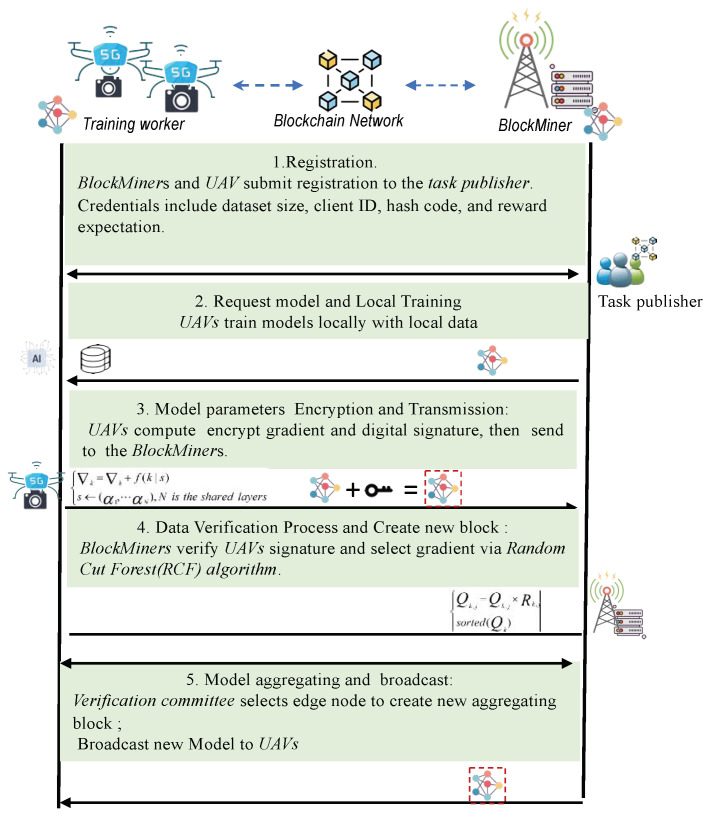
UBFL training process.

**Figure 5 sensors-24-01364-f005:**
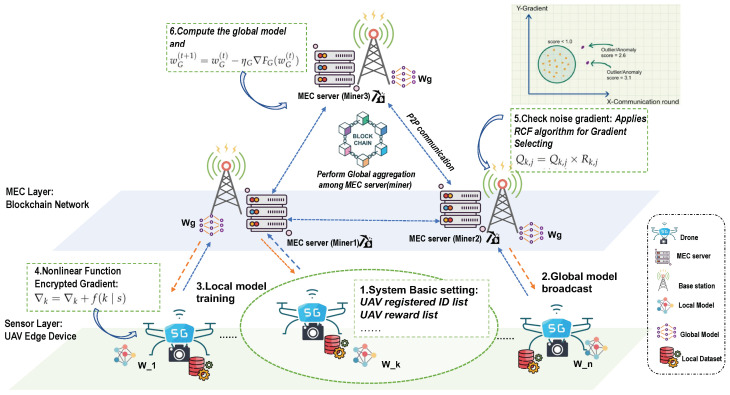
Identity authentication and gradient selection mechanism.

**Figure 6 sensors-24-01364-f006:**
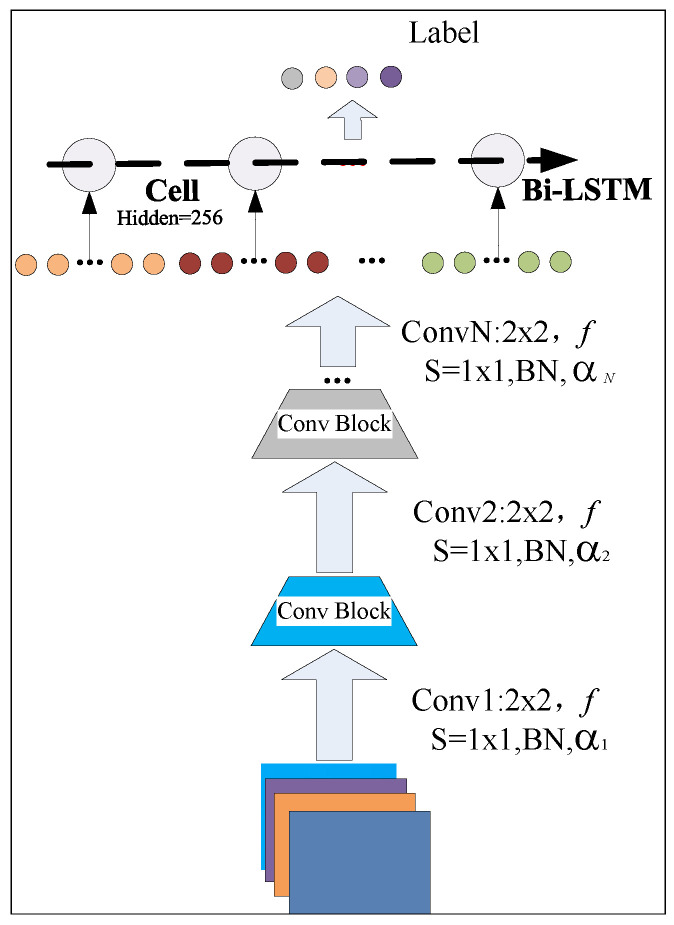
UAV local training neural network.

**Figure 7 sensors-24-01364-f007:**
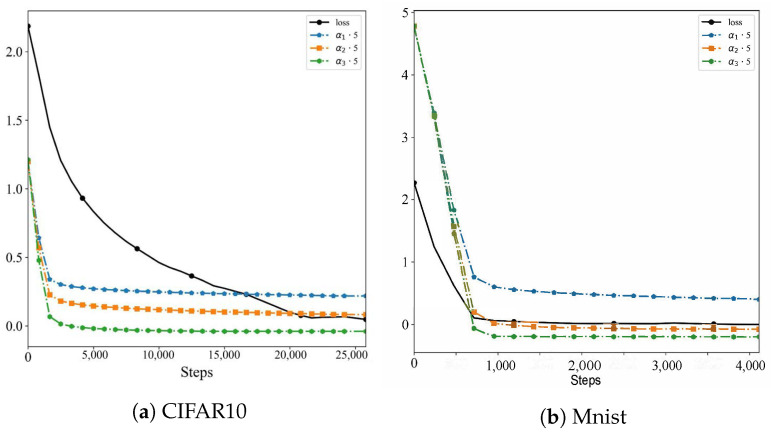
Optimization results of key parameters of UAV-BFL algorithm.

**Figure 8 sensors-24-01364-f008:**
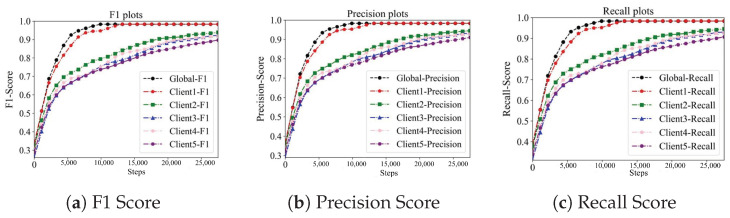
Impact of adaptive parameter on the learning performance in CIFAR10 dataset.

**Figure 9 sensors-24-01364-f009:**
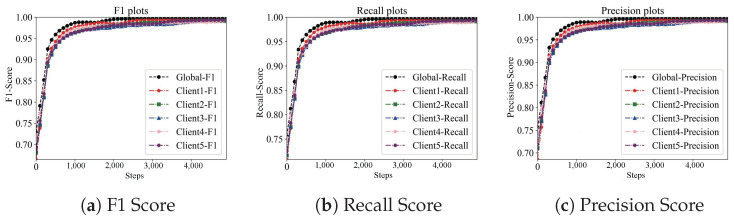
Impact of adaptive parameter on the learning performance in MNIST dataset.

**Figure 10 sensors-24-01364-f010:**
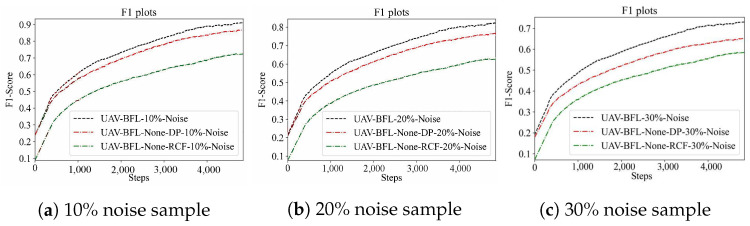
Impact of poison sample on the learning performance on the CIFAR10 dataset.

**Figure 11 sensors-24-01364-f011:**
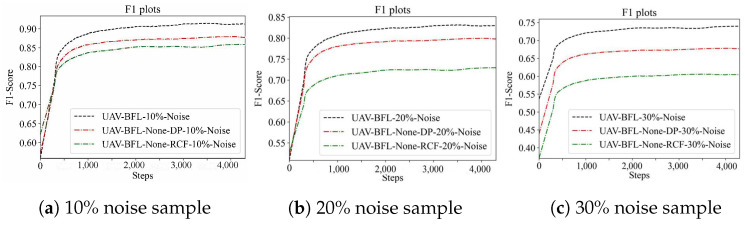
Impact of poison sample on the learning performance on the MNIST dataset.

**Table 1 sensors-24-01364-t001:** Allocation of experimental datasets.

Trainer	Dataset Size	Category	Train	Test
Global (blockchain)	—	—	—	—
$UAV_{1}$	‖ D ‖x$P_{1}$	C1		
$UAV_{2}$	‖ D ‖x$P_{2}$	C2		
$UAV_{3}$	‖ D ‖x$P_{3}$	C3	0.7	0.3
$UAV_{4}$	‖ D ‖x$P_{4}$	C4		
$UAV_{5}$	‖ D ‖x$P_{5}$	C5		

**Table 2 sensors-24-01364-t002:** Simulation parameter setting.

Batch Size	64
Learning rate	0.001
Truncation loss	100
Truncation loss	AdamOptimizer + Nonlinear adaptive parameters
UAV number	5
Shared network	The adaptive parameters of last three layers: $\alpha_{1}$, $\alpha_{2}$, $\alpha_{3}$

**Table 3 sensors-24-01364-t003:** Experiment hardware.

Parameter	Values
Computing power	RTX 2080Ti
operating system	Ubuntu18.04
Hard drive capacity	1000 GB
Number of CPU core	4
Number of GPU core	1∼6

**Table 4 sensors-24-01364-t004:** Experimental results of UAV-BFL loss and adaptive parameters.

Dataset	Parameter	Initial	Convergence
	UAV-BFL-Loss	2.11	0.080
	UAV-BFL-$\alpha_{1}$	1.0	0.043
Cifar10	UAV-BFL-$\alpha_{2}$	1.0	0.016
	UAV-BFL-$\alpha_{3}$	1.0	0.006
	UAV-BFL-Loss	0.77	0.004
	UAV-BFL-$\alpha_{1}$	1.0	0.047
Mnist	UAV-BFL-$\alpha_{2}$	1.0	0.019
	UAV-BFL-$\alpha_{3}$	1.0	0.007

**Table 5 sensors-24-01364-t005:** Comparison algorithms and evaluation indicators.

Experiment	Algorithm	Evaluation Indicators
Comparative experiment	UAV-BFL	Accuracy, F1-Score,
	UAV-BFL-Without-$\alpha_{1}$	Precision, Recall-Score
	UAV-BFL-Without-$\alpha_{2}$	
Comparative	UAV-BFL-Without-$\alpha_{3}$	
	UAV-BFL-DP	
	UAV-BFL-None-RCF	

**Table 6 sensors-24-01364-t006:** Global training accuracy on the Cifar10 and Mnist datasets.

Dataset	Algorithm	Accuracy	F1	Precision	Recall
	UAV-BFL	98.34	99.61	98.33	99.60
Cifar10	UAV-BFL-Without-$\alpha_{1}$	96.89	96.36	96.11	96.23
	UAV-BFL-Without-$\alpha_{2}$	97.30	96.22	96.67	97.11
	UAV-BFL-Without-$\alpha_{3}$	97.79	97.10	97.33	97.47
	UAV-BFL	99.60	99.89	99.88	99.91
Mnist	UAV-BFL-Without-$\alpha_{1}$	98.99	98.87	98.89	98.99
	UAV-BFL-Without-$\alpha_{2}$	99.19	98.99	98.99	98.99
	UAV-BFL-Without-$\alpha_{3}$	99.16	99.16	99.10	99.13

**Table 7 sensors-24-01364-t007:** Training accuracy compared on the Cifar10 and Mnist datasets.

Dataset	Algorithm	Accuracy	F1	Precision	Recall
Cifar10	UAV-BFL	98.34	99.61	98.33	99.60
	UAV-BFL-DP	96.34	95.47	96.00	96.10
Minist	UBFL	99.60	99.89	99.88	99.91
	UAV-BFL-DP	98.80	98.81	98.77	98.99
Performance Improvement ratio					
**Dataset**	**Algorithm**	**Accuracy**	**F1**	**Precision**	**Recall**
Cifar10	UAV-BFL	N/A	N/A	N/A	N/A
	UAV-BFL-DP	↑ 2.076	↑ 4.336	↑ 2.33	↑ 3.462
Mnist	UAV-BFL	N/A	N/A	N/A	N/A
	UAV-BFL-DP	↑ 0.810	↑ 0.903	↑ 1.124	↑ 0.929

N/A denotes the baseline. ↑ denotes the Improvement ratio for each baseline.

**Table 8 sensors-24-01364-t008:** RCF anomaly detection algorithm performance using F1 metric.

DataSet	Algorithm	10%	20%	30%
Cifar10	UAV-BFL	91.00	82.05	73.21
	UAV-BFL-None-DP	87.34	77.76	65.20
	UAV-BFL-None-RCF	72.12	63.44	59.77
Mnist	UAV-BFL	92.01	83.47	74.56
	UAV-BFL-None-DP	88.67	78.91	67.97
	UAV-BFL-None-RCF	84.10	74.57	60.44
Performance Improvement Ratio				
**DataSet**	**Algorithm**	**10%**	**20%**	**30%**
Cifar10	UAV-BFL	N/A	N/A	N/A
	UAV-BFL-None-DP	↑ 4.09%	↑ 5.57%	↑ 12.28%
	UAV-BFL-None-RCF	↑ 26.18%	↑ 4.29%	↑ 8.01%
Mnist	UAV-BFL	N/A	N/A	N/A
	UAV-BFL-None-DP	↑ 3.77%	↑ 5.77%	↑ 9.695%
	UAV-BFL-None-RCF	↑ 9.41%	↑ 11.94%	↑ 23.36%

N/A denotes the baseline. ↑ denotes the Improvement ratio for each baseline.

**Table 9 sensors-24-01364-t009:** RCF anomaly detection algorithm performance using Recall metric.

DataSet	Algorithm	10%	20%	30%
Cifar10	UAV-BFL	91.97	79.98	72.91
	UAV-BFL-None-DP	84.11	76.01	64.99
	UAV-BFL-None-RCF	71.67	62.75	59.29
Mnist	UAV-BFL	92.75	83.37	73.46
	UAV-BFL-None-DP	87.73	77.99	66.96
	UAV-UBFL-None-RCF	83.77	72.76	60.19
Performance Improvement				
**DataSet**	**Algorithm**	**10%**	**20%**	**30%**
Cifar10	UAV-BFL	N/A	N/A	N/A
	UAV-BFL-None-DP	↑ 9.34%	↑ 5.22%	↑ 12.19%
	UAV-BFL-None-RCF	↑ 28.32%	↑ 27.46%	↑ 22.97%
Mnist	UAV-BFL	N/A	N/A	N/A
	UAV-BFL-None-DP	↑ 5.72%	↑ 6.90%	↑ 9.71%
	UAV-BFL-None-RCF	↑ 10.72%	↑ 14.58%	↑ 22.05%

N/A denotes the baseline. ↑ denotes the Improvement ratio for each baseline.

## Data Availability

Data is available upon request due to restrictions related to development.

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
