# Peer review of "An Efficient Privacy Protection Mechanism for Blockchain-Based Federated Learning System in UAV-MEC Networks"

_sensors, 2024, doi:10.3390/s24051364_

Round 1

Reviewer 1 Report

Comments and Suggestions for Authors

Dear authors,

   The article entitled is interesting but it required major revision.

1.              Author should revise the grammatical and typo errors.

2.              The abstract needs to be revisited. For example, need to add motivation, data and significance based on the outcome.

3.              Author should add the main contributions in the introduction sections.

4.              Author should design the figs colourful and more interesting. 

                  5. Author should add the limitations of the study.

                  6. The conclusion needs to be updated with the key achievements and future scope of the work.

Comments on the Quality of English Language

Minor Changes required

Author Response

Thank you for your insightful feedback and constructive comments on our manuscript. We appreciate the opportunity to address the points you have raised, which have significantly contributed to improving the quality and depth of our research.

Please see the attachment, we provide responses to each of your comments.

Reviewer 2 Report

Comments and Suggestions for Authors

The paper presents an innovative approach to enhancing privacy protection within UAV-MEC networks through a blockchain-based federated learning strategy, termed UBFL. It addresses the limitations of traditional UAV-assisted MEC systems by incorporating blockchain technology to establish a decentralized architecture for secure model updates and data validation without a central server. The key contributions include an adaptive nonlinear function-based algorithm to safeguard UAV model updates' privacy and the use of Random Cut Forest (RCF) for anomaly detection to counteract poisoning data attacks, demonstrating superior privacy and security over traditional methods.

Pros:

1. The integration of blockchain technology into UAV-MEC networks for privacy protection represents a forward-thinking approach to addressing security vulnerabilities. 

2. The adaptive nonlinear function-based algorithm and the employment of RCF for anomaly detection are innovative solutions for enhancing data privacy and integrity.

3. The paper demonstrates the effectiveness of its proposed solutions through extensive experiments on CIFAR10 and MNIST datasets, showing improved accuracy and resistance to data poisoning attacks.

Cons:

1. The paper did not fully address the computational complexity and scalability of the proposed blockchain-based approach, especially in large-scale UAV networks. 

2. Although the paper claims superiority over traditional methods, a more detailed comparison with a wider range of existing solutions could strengthen the argument.

3. The real-world applicability of the proposed system, considering the dynamic and potentially constrained environment of UAVs, requires further elaboration.

In conclusion, the paper introduces a promising blockchain-based federated learning framework that significantly enhances privacy and security in UAV-MEC networks. Its novel use of an adaptive nonlinear function and RCF for anomaly detection represents a significant step forward in addressing the challenges of privacy preservation and security in these systems. However, further research into its scalability, detailed comparisons with existing methods, and the practical deployment challenges would provide a more comprehensive understanding of its potential impact. The study's contributions to the fields of UAV-enabled computing and federated learning are commendable, indicating a positive direction for future research in secure and efficient UAV-MEC network operations.

Decision: Recommend for acceptance with minor revisions.

Author Response

(The authors gave the same response as above.)

Reviewer 3 Report

Comments and Suggestions for Authors

This paper proposes a good piece of work in the domain, however, some concerns about the paper are mentioned as follows:

1. Authors should include a paragraph in Introduction outlining the structure of the paper.

2. The details of calculating the global model parameter as shown in equations 3 and 4 are not enough. Authors should show how the global model parameters are calculated.

3. The use of MINST and CIFAR10 datasets is not convincing. Since the domain is UAV, how come these kinds of data reflect the syntax of such application? Authors should use a dataset such that it reflects the context. Perhaps, the use of synthetic dataset generated in the same UAV context will be more convincing.

4. Since authors proposed a method for detecting anomalies using RCF, they are required to compare the results of such technique with other previously used unsupervised techniques in the future.

Comments on the Quality of English Language

English is good. Some minor editings are required.

Author Response

(The authors gave the same response as above.)

Round 2

Reviewer 1 Report

Comments and Suggestions for Authors

1. Extensive English editing is required. 

2. The title is quite long and could be made more concise without losing its essential meaning.  3. Some issues need to be corrected in the abstract and introduction 4. Insufficient Background Information 5. Some grammatical errors to be found in page 3 and paragraph 2. it needs to be corrected.

Author Response

(The authors gave the same response as above.)
